# *Leishmania braziliensis* causing human disease in Northeast Brazil presents *loci* with genotypes in long-term equilibrium

Juliana A. Silva[1,2], Ana Isabelle Pinheiro[1,2], Maria Luiza Dourado[1], Lilian Medina[1,2], Adriano Queiroz[3], Luiz Henrique Guimarães[1,4,5], Marcus Miranda Lessa[2], Ednaldo L. Lago[1,4], Paulo Roberto L. Machado[1,2,4], Mary E. Wilson[6], Edgar M. Carvalho[1,2,3,4], Albert Schriefer[1,2,4,7]*

**1** Serviço de Imunologia, Hospital Universitário Professor Edgard Santos, Universidade Federal da Bahia (UFBA), Salvador, Brazil, **2** Programa de Pós-graduação em Ciências da Saúde, Faculdade de Medicina da Bahia, UFBA, Salvador, Brazil, **3** Instituto Gonçalo Moniz, Fundação Instituto Oswaldo Cruz, Salvador, Brazil, **4** Instituto Nacional de Ciência e Tecnologia em Doenças Tropicais (INCT-DT), Salvador, Brazil, **5** Universidade Federal do Sul da Bahia, Teixeira de Freitas, Brazil, **6** Departments of Internal Medicine and Microbiology, University of Iowa and the VA Medical Center, Iowa City, Iowa, United States of America, **7** Departamento de Ciências da Biointeração, Instituto de Ciências da Saúde, UFBA, Salvador, Brazil

* nab.schriefer@gmail.com

**Data Availability Statement:** Sequencing data of the entire dataset have been posted in Figshare.

## Abstract

### Background

Leishmaniases are neglected tropical diseases that inflict great burden to poor areas of the globe. Intense research has aimed to identify parasite genetic signatures predictive of infection outcomes. Consistency of diagnostic tools based on these markers would greatly benefit from accurate understanding of *Leishmania* spp. population genetics. We explored two chromosomal *loci* to characterize a population of *L. braziliensis* causing human disease in Northeast Brazil.

### Methodology/Principal findings

Two temporally distinct samples of *L. braziliensis* were obtained from patients attending the leishmaniasis clinic at the village of Corte de Pedra: (2008–2011) primary sample, N = 120; (1999–2001) validation sample, N = 35. Parasites were genotyped by Sanger's sequencing of two 600 base pairs *loci* starting at nucleotide positions 3,074 and 425,451 of chromosomes 24 and 28, respectively. Genotypes based on haplotypes of biallelic positions in each *locus* were tested for several population genetic parameters as well as for geographic clustering within the region. Ample geographic overlap of genotypes at the two *loci* was observed as indicated by non-significant Cusick and Edward's comparisons. No linkage disequilibrium was detected among combinations of haplotypes for both parasite samples. Homozygous and heterozygous genotypes displayed Hardy-Weinberg equilibrium (HWE) at both *loci* in the two samples when straight observed and expected counts were compared by Chi-square (p>0.5). However, Bayesian statistics using one million Monte-Carlo randomizations disclosed a less robust HWE for chromosome 24 genotypes, particularly in the

com under DOI https://doi.org/10.6084/m9.
figshare.19601473.v1.

**Funding:** This work was supported in part by US
National Institutes of Health (NIH) grants AI136862
and U01-AI136032 (EMC, PRLM, AS), and by
grants from the US Department of Veterans' Affairs
I01BX001983 and I01BX000536 (MEW). JAS, AIP
and LM were recipients of Coordenação de
Aperfeiçoamento de Pessoal de Nível Superior -
Brazil (CAPES) PhD (JAS, LM) or MS scholarships
(AIP). MLD was the recipient of a Conselho
Nacional de Desenvolvimento Científico e
Tecnológico - Brazil (CNPq) scientific initiation
scholarship. Funders played no role in the study
design, data collection and analysis, decision to
publish, or preparation of the manuscript.

**Competing interests:** The authors have declared
that no competing interests exist.

primary sample (p = 0.04). Fixation indices (Fst) were consistently lower than 0.05 among
individuals of the two samples at both tested *loci*, and no intra-populational structuralization
could be detected using STRUCTURE software.

## Conclusions/Significance

These findings suggest that *L. braziliensis* can maintain stable populations in foci of human
leishmaniasis and are capable of robust genetic recombination possibly due to events of
sexual reproduction during the parasite's lifecycle.

## Author summary

*Leishmania braziliensis* affects poor human populations in the tropics, may cause face dis-
figuring lesions and may also resist treatment. There has been intense research for mark-
ers in these parasites genetic contents for helping predict if an infected human being
would be of greater chance of severe disease or treatment failure. The consistent identifi-
cation of such markers requires a deep understanding of how genes circulate within these
parasites' natural populations. We explored two small segments of DNA (i.e. *loci*), one on
chromosome 24, the other on chromosome 28 of *L. braziliensis* to characterize a popula-
tion that causes human disease in Northeast Brazil. We employed two samples of parasites
obtained from lesions of patients diagnosed from 1999 to 2001, and from 2008 to 2011.
We sequenced the DNA of those *loci* in each parasite of the two samples. Then, we evalu-
ated the status of several population genetics parameters among them. Based on our find-
ings to that region, we concluded that *L. braziliensis* can maintain populations that are
genetically stable for several years in foci of human leishmaniasis and are capable of robust
recombination of their genetic contents, probably due to events of sexual reproduction
during its lifecycle.

## Introduction

Leishmaniases are vector-borne diseases that present an overall incidence of close to 1.5 mil-
lion new cases per year [1], and cause an approximate burden of 2 million disease adjusted life
years (DALYs) lost [2] in tropical and subtropical areas of the globe. In the Americas, endemic
foci distribute from Mexico to Argentina and patients may present visceral or tegumentary
disorders. American tegumentary leishmaniasis (ATL) is pleomorphic with novel clinical pre-
sentations continuously emerging in affected regions [3,4]. Emerging forms of ATL have
tended to be refractory to antimony therapy, requiring timely switch to second line treatment
to avoid more severe, sometimes disfiguring disease [3,4]. For this reason, several groups have
attempted to identify parasite markers or biological signatures that associate with ATL out-
come [5–9].

  To develop consistent predictive diagnostic tools based on genomic markers, proper under-
standing of *Leishmania* spp. populations is necessary. Early evidence originally led to the
assumption that *Leishmania* spp. reproduced mostly asexually, resulting in structured popula-
tions made of several different clonal strains of the protozoa [10]. More recent work has
strongly shaken this paradigm. Two complementary experimental studies showed that *Leish-
mania* parasites with different genetic backgrounds are capable of sexual reproduction within
sand flies, resulting in offsprings that are hybrids of parental genotypes [11,12]. If this

observation proved widespread in natural foci of leishmaniasis, it would affect the way parasite diagnostic markers are screened for, and also help shed light into why novel forms of ATL may emerge so fast and become stable features of the clinical landscape in those regions.

Corte de Pedra is a region hyperendemic for ATL secondary to *Leishmania (Viannia) braziliensis* infection in Northeast Brazil. *L. braziliensis* causes four distinct forms of ATL in this region: localized cutaneous (CL) and mucosal leishmaniasis (ML) are traditional forms; while disseminated (DL) and atypical cutaneous leishmaniasis (ACL) have been emerging in Corte de Pedra [3,4,13,14]. Underlying this variety of clinical presentations is a complex population of the parasite [15]. In one study, we identified several small genomic *loci* distributed among different chromosomes of the protozoa that presented two or more haplotypes of polymorphic nucleotides in that population of *L. braziliensis* [5]. We have ever since explored these polymorphic *loci* to better understand that population and to address whether *L. braziliensis* genotype is an important determinant of ATL outcome [5,6].

In the current study, we explored two of the described *loci* to investigate whether genetic recombination suggestive of sexual reproduction is frequent in the *L. braziliensis* of Corte de Pedra. Both *loci* are approximately 600 base pairs long. One genomic *locus* starts at nucleotide position 425,451 on chromosome 28 (referred to as CHR28/425451) while the second *locus* starts at position 3,074 on chromosome 24 (i.e. CHR24/3074) [5]. We tested whether genetic recombination was frequent assessing haplotype frequencies for Hardy-Weinberg equilibrium (HWE) at both *loci*. Long-term stability of that *L. braziliensis* population was verified by calculating the fixation index (Fst) between two samples of parasite isolates obtained from ATL patients of Corte de Pedra diagnosed approximately ten years apart.

## Methods

### Ethics statement

This study was approved by the Institutional Review Board of the School of Medicine of the Federal University of Bahia, Brazil (IRB approval reference number CAAE–3041.0.000.054.07). Written consent was obtained from all participating subjects.

### Study area

The study area is a hyper-endemic focus for ATL caused by *L. braziliensis*. This region has been called Corte de Pedra because the major leishmaniasis clinic employed in the enrollment of participants for several studies is located in a village formerly known by this name. Corte de Pedra is composed of 20 municipalities in a rural area located in the southeastern region of the state of Bahia, in the northeast of Brazil, falling within geographic coordinates (latitude / longitude) -14˚/-39˚, -13˚/-39˚, -14˚/-40˚, -13˚/-40˚. Residents of this area work mostly in agriculture, which is often carried out in primary or secondary forests [16,17].

### Patients and disease definitions

All subjects in the study resided in the *L. braziliensis* endemic region, were self-referred to and were diagnosed at the leishmaniasis clinic in Corte de Pedra. The mean time of study participants residences at their current addresses during diagnosis and parasite sampling was 17 years (more precisely: mean = 17.6 years; median = 18.0 years; minimum time of residence = 1.0 year; maximum time of residence = 70.0 years). More than 90% of the study participants lived on farms. Clinical criteria for CL included fewer than 10 ulcerative skin lesions without evidence of mucosal involvement. DL was defined as a disease with more than 10 acneiform, papular or ulcerative skin lesions spread over 2 or more body areas, with or without

mucosal involvement. ML was defined by metastatic mucosal lesions affecting the nose, palate, pharynx, or larynx but not contiguous with primary cutaneous lesions, with or without primary skin lesions compatible with those of CL. Besides other standard ATL diagnosis procedures, all subjects had infection confirmed by parasite DNA detection in lesion biopsy specimens by real-time PCR [18].

## Study samples

Two distinct samples were employed in the research. The primary sample was obtained from subjects enrolled between August 2008 and July 2011. It consisted of *L. braziliensis* drawn from 336 subjects recruited according to the criteria of roughly two CL per each ML or DL case, matched for month of diagnosis. The included participants consisted of 35 patients with ML, 76 with DL, and 225 with CL. Almost all ML and DL patients attending the leishmaniasis clinic during that period were enrolled in the study. We included double the number of CL subjects relative to ML and DL in order to partially account for the fact that CL is much more frequent in Corte de Pedra. This recruitment yielded a sample of 120 parasite isolates, which originated from 87 CL, 24 DL and 9 ML patients.

The validation sample was used to confirm some of the findings obtained with the primary sample. It was a convenience sample of 35 parasites obtained from ATL patients diagnosed in the same leishmaniasis clinic between 1999 and 2001, under the same procedures as the primary sample. Those parasites were isolated from 17 CL, 9 DL and 9 ML patients.

Of the 120 isolates of *L. braziliensis* obtained from ATL patients in the 2008–2011 sample, 111 had adequate genotyping data at *locus* CHR24/3074 and 117 at *locus* CHR28/425451 (i.e. presented four to six clones per parasite isolate with at least 600 base pairs of readable sequence) to be used in the performed population genetics tests. In the validation sample of 1999–2001 all thirty-five *L. braziliensis* isolates presented four to six clones with adequate sequence at both *loci*.

## Parasites

The *L. braziliensis* isolates used in the study were cultured from aspirates of the borders of skin and mucosal lesions. Aspirated material was immediately suspended in biphasic liver infusion tryptose/Novy, McNeal, Nicolle (LIT/NNN) medium and incubated at 26˚C for two to three weeks. The suspension was then transferred to complete Schneider's medium supplemented with 10% heat-inactivated fetal calf serum and Gentamicin 50 mg/mL (Sigma-Aldrich) and incubated at 26˚C for up to another two weeks. Parasite stocks were frozen without further subculture in 10% DMSO / 90% growth medium in liquid nitrogen. All DNA used in the study was extracted from freshly isolated parasites in the primary sample, collected between 2008 and 2011, and from first thaw from first passaged frozen isolates in the validation sample, collected between 1999 and 2001. This strategy aimed to avoid the accumulation of polymorphisms in the genomes of isolates undergoing multiple culture passages. The isolates DNA were employed for species and genotype determination as described below.

## Genomic DNA extraction and parasite species determination by PCR

Genomic DNA was extracted from suspensions containing approximately $10^6$ promastigotes as previously described [5,15] then resuspended in 100μL of TE (Tris-HCl 10mM, EDTA 1mM pH 8.0) buffer. Long-term storage DNA aliquots were kept at –70˚C, while test samples were maintained at –20˚C until used. Leishmania species was determined by a serial real-time quantitative PCR assay system [18]. All parasite isolates used in the study had the species thus confirmed as *L. braziliensis*.

## Genotyping *L. braziliensis* isolates

Parasites were genotyped according to the haplotypes of polymorphic nucleotides contents in the *loci* CHR28/425451 and CHR24/3074, previously shown to distinguish *L. braziliensis* strains of Corte de Pedra [5]. These markers derive from two previous studies in which we began to characterize the *L. braziliensis* population endemic in the study region at Northeast Brazil [5, 15]. In summary, we randomly amplified, cloned, sequenced and compared several regions of the parasites' genomes. Then, we searched for haplotypes of single nucleotide polymorphisms that could reproducibly discriminate strains among individuals of that population. We detected six loci presenting such haplotypes that distributed over five different chromosomes of *L. braziliensis* [5].

In the current research, we opted to use markers of that panel that fell within previously annotated genes located in two different chromosomes of the parasite. They marked the putative coding regions annotated for Pyruvate Dehydrogenase Kinase (CHR24/3074 haplotypes) and for Long-chain Fatty Acid coenzyme A Ligase (CHR28/425451 haplotypes). In this way, we attempted to partly mitigate potentially negative consequences of using anonymous markers, like the ones that fell on chromosomes 26, 32 and 35 in the previous report [5]. One such consequence we tried to overcome was a high mutation rate that would cause the continuous emergence of new haplotypes in the population if the marker happened to fall on a non-coding region of the parasite's genome.

Primers 5´:TAAGGTGAACAAGAAGAATC plus 5´:CTGCTCGCTTGCTTTC were used to amplify a 622 nucleotide-long segment in CHR28/425451, and 5':GGACTGGAGTGATC-GAA plus 5':TGGCTCAAGTGTCGCA to amplify a 779 nucleotides fragment in CHR24/3074 from parasite genomic DNA as previously described [5]. Amplicons were cloned using the Original TA Cloning Kit pCR 2.1 VECTOR (Invitrogen, Thermo Fisher Scientific Co., MA, USA), according to manufacturer's instructions. Briefly, the amplicons were inserted by overnight ligation into PCR 2.1 plasmids, which were used for chemical transformation of competent DH5α *Escherichia coli*. Plasmid minipreps were generated from six recombinant bacteria colonies per study isolate. Amplicon cloning was confirmed by digestion analysis, using Eco RI restriction endonuclease (Invitrogen Inc.). Plasmid inserts were sequenced by the Sanger method with primers complementary to the M13 vector sequences. Sequencing was performed at Macrogen Inc. (Seoul, South Korea). Mega X software [19] was used to align the sequences at each of the two test *loci* in order to determine the SNP/indel haplotypes detectable in each study isolate.

## Mapping *L. braziliensis* genotypes in the study area

High-resolution distribution of ATL patients that yielded genotyped *L. braziliensis* isolates to the study was determined by acquisition of geographic coordinates using the GPS device Garmin GSX 60 (Garmin, Riverton, WY, USA). Because leishmaniasis is believed to be transmitted mostly within plantations where most residents of the region live and work, patients' residences were used as reference points for standardization purposes. The data was plotted for visual inspection onto a high-definition satellite photograph of Corte de Pedra region (ENGESAT, Curitiba, Brazil) using ArcGis version 10 software (Environmental Systems Research Institute Inc., Redlands, CA, USA). We compared the geographic distribution of different groups of *L. braziliensis* genotypes in Corte de Pedra using the Cuzick and Edward's test in the geostatistical package Clusterseer version 2.2.4 (Terraseer Inc., Ann Arbor, MI, USA). This test results significant when two groups of geographic events distribute differently over the study area.

## Testing for Hardy-Weinberg equilibrium (HWE) and linkage disequilibrium (LD) of *L. braziliensis* genotypes

We tested homozygous and heterozygous genotypes at *loci* CHR28/425451 and CHR24/3074 for HWE in order to address whether genetic recombination among the individuals in the *L. braziliensis* population of Corte de Pedra was vigorous, and suggestive of potential occurrence of sexual reproduction. HWE is tested by comparing observed counts of homozygous and heterozygous genotypes at a given *locus* among the individuals in the study sample versus the counts that should be expected if those genotypes were in equilibrium based on the equation $p^2 + 2pq + q^2 = 1$. Observed counts and expected counts of homozygous and heterozygous genotypes are then compared by chi-square as a goodness of fit test. In comparisons that rendered non-significant by chi-square (i.e. $p > 0.05$) with this approach, we considered the genotypes to be in equilibrium within the target *L. braziliensis* population.

In order to assess the robustness of the equilibrium detected among genotypes in the two *loci* for the *L. braziliensis* samples with the strategy above, we further tested them for HWE employing Markovian Chain Monte Carlo (MCMC) simulations with the Arlequin 3.5.2.2 population genetics software [20]. In these analyses, $10^6$ MCMC iterations were run after a burn-in step of $10^5$ simulations. A $p < 0.05$ was considered significant.

LD between CHR24/3074 and CHR28/425451 haplotypes was tested within each sample employing only those *L. braziliensis* isolates that presented genotyping information for both *loci*: N = 35 for 1999–2001, N = 103 for 2008–2011. The significances of the exact tests for LD between the polymorphic positions at the two *loci* were calculated after 10,000 permutations in Arlequin 3.5.2.2. A $p < 0.05$ was considered significant.

## Determining heterozygozity indices ($H_T$, $H_S$) and level of differentiation (Fst) between primary (2008–2011) and validation samples (1999–2001) of *L. braziliensis*

Heterozygozity (H) consists in the observed ($H_O$) and expected ($H_E$) frequencies of heterozygotes for a given *locus* among individuals of a population. The observed frequencies of heterozygotes $H_O$ was calculated simply by dividing the actual counts of heterozygotes in the target *locus* (in our study: CTTCAG:TCATGA at CHR24/3074 and CCT:TT- at CHR28/425451) per N, that is, the size of the sample (i.e. N = 35 for both *loci* in the 1999–2001 sample; N = 111 at CHR24/3074 and N = 117 at CHR28/425451 in the 2008–2011 sample). The expected frequencies $H_E$ consisted in 2pq calculated for each sample based on the observed frequencies of each allele during HWE testing (i.e. p and q, in the current study consisting of: CTTCAG and TCATGA at CHR24/3074; CCT and TT- at CHR28/425451).

In order to calculate the fixation index Fst, the frequencies of the involved alleles over all population (i.e. the 2008–2011 and 1999–2001 samples of the study) p-bar and q-bar were first determined, according to the following equations, using the *locus* CHR28/425451 as the example: p-bar = (Counts of CCT in 2008–2011 sample + Counts of CCT in 1999–2001 sample) / ($2 \times N_{2008-2011} + 2 \times N_{1999-2001}$); q-bar = (Counts of TT- in 2008–2011 sample + Counts of TT- in 1999–2001 sample)/($2 \times N_{2008-2011} + 2 \times N_{1999-2001}$). The same parameters were calculated for CHR24/3074, employing the alleles CTTCAG and TCATGA. Finally, for each of the two *loci* the fixation index was calculated as Fst = $(H_T - H_S) / H_T$. In this equation, $H_T = (2 \times \text{p-bar} \times \text{q-bar})$ and HS = ($H_{E2008-2011} \times N_{2008-2011} + H_{E1999-2001} \times N_{1999-2001}$) / ($N_{2008-2011} + N_{1999-2001}$).

### Identifying structuralization in the *L. braziliensis* population

Structuralization within the *L. braziliensis* population into discrete subpopulations in Corte de Pedra was assessed using the software STRUCTURE version 2.3.4. [21,22]. The analyses were performed using data of CHR24/3074 and CHR28/425451 *loci* simultaneously. All isolates that were used for HWE and Fst analyses in the 1999–2001 and 2008–2011 samples were employed. Two conditions were tested: structuralization into two subpopulations (i.e. K = 2), structuralization into 20 subpopulations (i.e. K = 20). The proportional contributions of each individual in the overall sample to each theoretical subpopulation were graphed in histograms ordered by sampling period (i.e. 1999–2001 and 2008–2011) after a burn-in step of 10,000 runs followed by 100,000 MCMC replications.

## Results

### Study samples time distributions

All 155 isolates used in the study were confirmed as *L. braziliensis* by qPCR. Parasites were obtained from ATL subjects during monthly visits to the endemic site in Corte de Pedra, Northeast Brazil. In the primary sample the enrollment occurred during 6 consecutive semesters, corresponding to six transmission seasons, between July/2008 and June/2011. The amounts of isolates obtained were fairly similar across transmission seasons, with an average 15 isolates per semester, except for a limited surge during 2009 (Fig 1A). The validation sample consisted of 35 isolates drawn from ATL patients of the same region between July/1999 and December/2001. The time distribution of its isolates was more irregular than that of the primary sample, with approximately half of the parasites obtained in the second semester of 2000 and first semester of 2001 (Fig 1B).

### Predominant haplotypes in the *L. braziliensis* population of Corte de Pedra

Multiple alignment of DNA sequences across the panel of *L. braziliensis* permitted identify consistent haplotypes of nucleotides in biallelic polymorphic positions 31, 52, 68, 129, 469 and 604 in *locus* CHR24/3074, and at 30, 286 and 545 in CHR28/425451. Each *locus* presented with two clades of haplotypes in the parasite population. Eight different haplotypes could be detected in CHR24/3074 (Fig 2A), while CHR28/425451 presented only three (Fig 2B). Despite this diversity, only two haplotypes per *locus*, belonging to different clades, could be found in at least 20% of the study isolates: CTTCAG and TCATGA in CHR24/3074; CCT and TT- in CHR28/425451 (Fig 2).

### *L. braziliensis* genotypes in primary and validation study samples

Focusing on the four haplotypes that crossed the 20% frequency cut-off, we assembled genotypes for the isolates in the samples based on their haplotype contents at each *locus*, assuming a diploid genome. Overall, the observed frequencies for the CHR24/3074 genotypes CTTCAG: CTTCAG, CTTCAG:TCATGA and TCATGA:TCATGA consisted of approximately 54%, 44% and 2% in the primary sample, and of 77%, 20% and 3% in the validation sample, respectively (Table 1). CHR28/425451 CCT:CCT, CCT:TT- and TT-:TT- observed frequencies consisted of 29%, 58% and 13% in the primary sample, and of 20%, 49% and 31% in the validation sample, respectively (Table 1).

Time distributions of homozygous and heterozygous genotypes in CHR24/3074 and CHR28/425451 displayed overall semester-to-semester fluctuations in their proportions for both *loci* in the two study samples (Fig 3), possibly as the result of the random success of

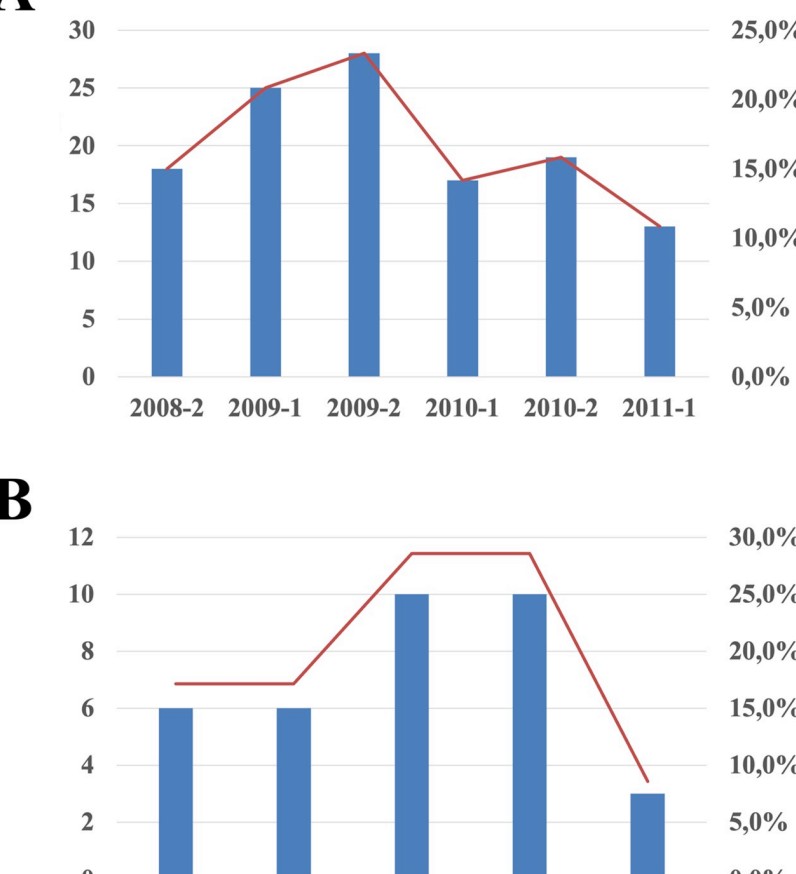

**Fig 1. Time distributions of *L. braziliensis* isolations for the primary (2008–2011) and validation (1999–2001) parasite samples used in the study.** (A) Primary sample of parasite isolates obtained from ATL patients enrolled during six consecutive semesters from July/2008 to June/2011 in Corte de Pedra, Northeast Brazil. (B) Validation sample obtained from ATL patients enrolled during five consecutive semesters from July/1999 to December/2001 in the same region. Columns depict the number of isolates obtained per semester, according to the left Y axes. Lines consist in percent contribution of each semester to the total number of isolates in the corresponding sample, according to right Y axes.

different strains of the parasite in reaching the human population of the affected focus at each transmission season. However, the fluctuation was not statistically significant (Chi-square p>0.05).

## Hardy-Weinberg Equilibrium and linkage disequilibrium of CHR24/3074 and CHR28/425451 genotypes

In order to infer whether genetic recombination is a common event in the *L. braziliensis* population of the study region, we tested if genotypes at CHR24/3074 and CHR28/425451 were in HWE in the primary sample, and then confronted the observations with those for the

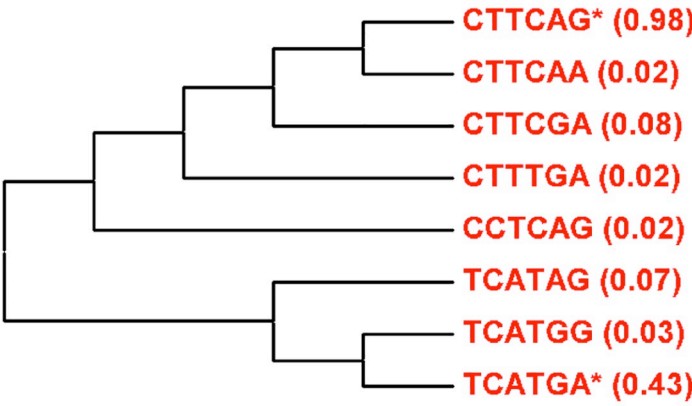

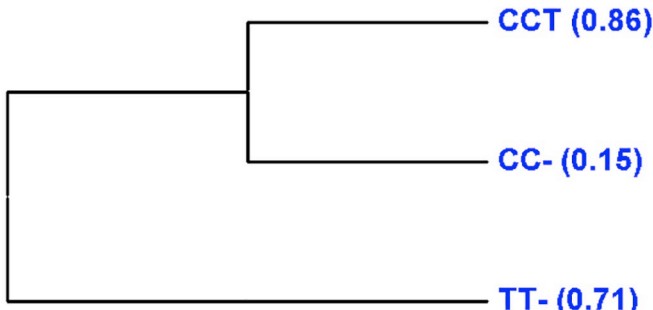

**Fig 2. Haplotypes of nucleotides found in biallelic polymorphic positions at chromosomal *loci* CHR24/3074 and CHR28/425451 of at least two different isolates in the collection of *L. braziliensis* used in the study.** For each *locus*, DNA sequences of four to six clones per *L. braziliensis* isolate of the study collection (i.e. primary plus validation samples) were aligned using MEGA X. Multiple alignment across all study isolates permitted identify polymorphic nucleotides at biallelic positions 31, 52, 68, 129, 469 and 604 in *locus* CHR24/3074, and 30, 286 and 545 in CHR28/425451. (A) Haplotypes of nucleotides detected within polymorphic positions at CHR24/3074; (B) Haplotypes of polymorphic nucleotides detected in CHR28/425451. The proportional representation of each haplotype in the sample is within parenthesis at the tips of the dendrograms.

validation sample. Observed and expected counts of homozygous and heterozygous genotypes for the two *loci* in each parasite sample can be found in Table 1. Straight comparison of observed versus expected counts using Chi-square as the goodness of fit test rendered non-significant, indicating that the genotypes in the tested *loci* are in HWE in the parasites of Corte de Pedra (Table 1). In order to inquire on the strength of the detected equilibrium, we repeated

**Table 1. Hardy-Weinberg equilibrium testing of genotypes in two temporally distinct *L. braziliensis* populations endemic in Corte de Pedra, Northeast Brasil: 1999–2001 and 2008–2011.**

| Population | *Locus* | Genotype | Observed counts[*] | Expected counts[#] | HWE p-value[&] | HWE/MCMC p-value[@] |
|---|---|---|---|---|---|---|
| **1999–2001** | CHR24/3074 (n = 35) | CTTCAG:CTTCAG | 27 | 27 | 0.97 | 0.44 |
| | | CTTCAG:TCATGA | 7 | 8 | | |
| | | TCATGA:TCATGA | 1 | 1 | | |
| | CHR28/425451 (n = 35) | CCT:CCT | 7 | 7 | 1.00 | 1.00 |
| | | CCT:TT- | 17 | 17 | | |
| | | TT-:TT- | 11 | 11 | | |
| **2008–2011** | CHR24/3074 (n = 111) | CTTCAG:CTTCAG | 60 | 64 | 0.22 | 0.04 |
| | | CTTCAG:TCATGA | 49 | 40 | | |
| | | TCATGA:TCATGA | 2 | 6 | | |
| | CHR28/425451 (n = 117) | CCT:CCT | 34 | 40 | 0.30 | 0.06 |
| | | CCT:TT- | 68 | 57 | | |
| | | TT-:TT- | 15 | 21 | | |

[*] Genotype counts actually observed in the test sample.

[#] Genotype counts expected to be found in the sample if genotypes were in HWE.

[&] p-value of the Chi-square comparison between observed and expected genotypes counts.

[@] p-value of the Chi-square comparisons between observed and expected genotypes counts obtained during one million Markovian Chain Monte-Carlo simulations of the data, preceded by 100,000 burn-in simulations.

the comparisons using Bayesian statistics with one million Markovian Chain Monte-Carlo simulations of the data. MCMC rendered non-significant and reinforced the original HWE findings for CHR28/425451, but also displayed that equilibrium at CHR24/3074 was less robust, particularly for parasites of the 2008–2011 sample.

Further suggesting that genetic recombination within that natural population of *L. braziliensis* is frequent, CHR24/3074 polymorphisms did not show linkage disequilibrium with those of CHR28/42451, as indicated by pair-wise comparison values above 0.05 in Table 2. As expected, sets of polymorphic nucleotides within each *locus* were in LD, displaying pair-wise comparison values below 0.05 in Table 2, in accordance with our option of using haplotypes defined by biallelic positions at each *locus* instead of just alleles in individual polymorphic positions in our population analyses. Nevertheless, LD was not detected between pairs of haplotypes simultaneously occurring at each of the two *loci*. This observation suggests the occurrence of independent segregation of haplotypes found in different chromosomes of *L. braziliensis* in the population, in contrast to what might be expected if the mode of reproduction in these parasites were predominantly clonal.

## Spatial distribution of *L. braziliensis* genotypes in the focus of endemicity

The findings of the HWE analyses necessarily imply the existence of geographic opportunity for the parasite strains to exchange the involved haplotypes. We tested for this by high-resolution mapping the *L. braziliensis* isolates in each of the two study samples onto a satellite photograph of the endemic region (Fig 4), and then assessing their overlap with the geostatistical test of Cusick and Edward. Homozygous and heterozygous genotypes for each of the two tested *loci* (i.e. CHR24/3074, Fig 4A and 4C; CHR28/425451, Fig 4B and 4D) and in both time frames of analysis (i.e. 1999–2001, Fig 4A and 4B; 2008–2011, Fig 4C and 4D) presented broad distributions, not showing any tendency to segregate into spatial clusters within Corte de Pedra.

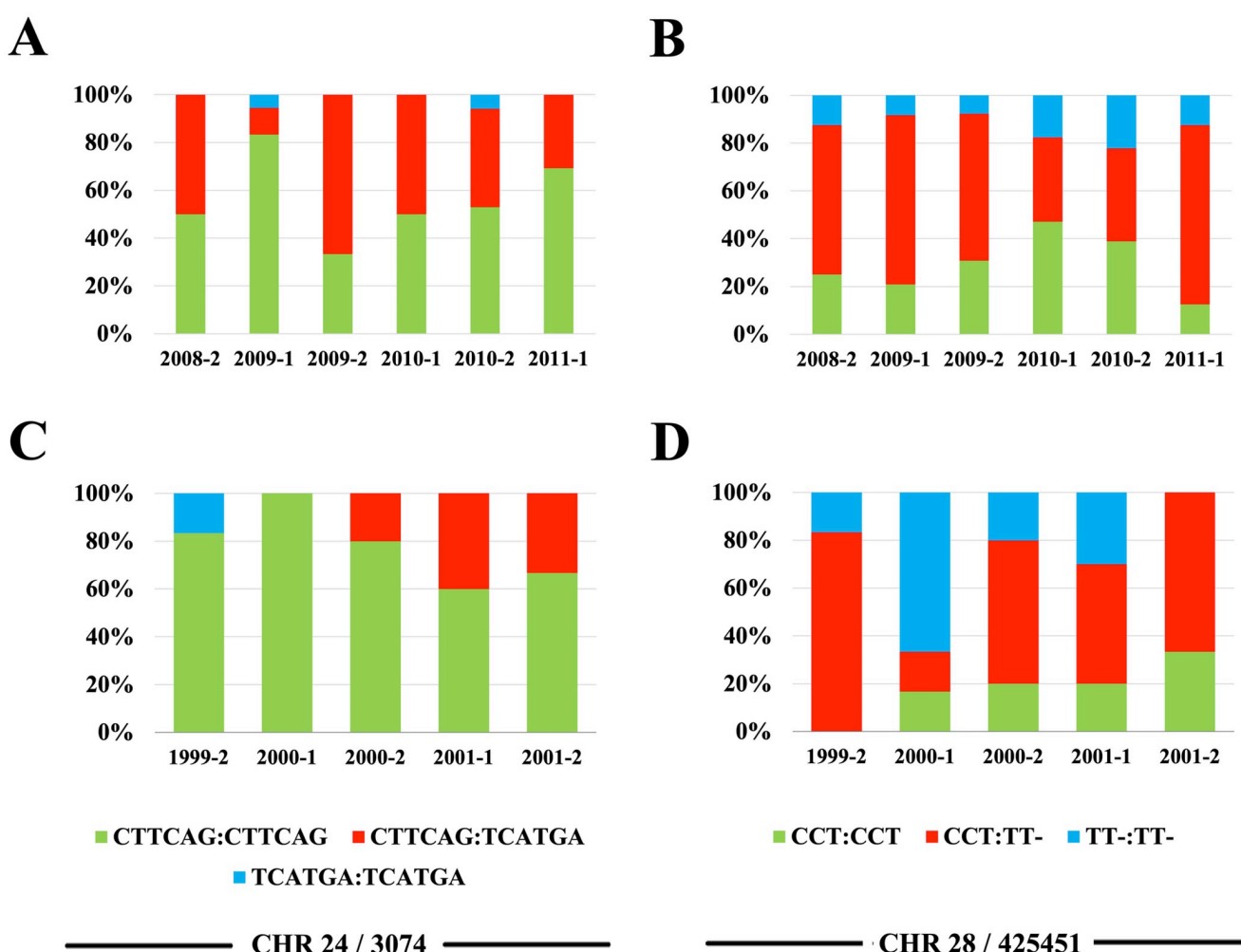

**Fig 3. Time distributions of homozygous and heterozygous genotypes in CHR24/3074 and CHR28/425451.** Each *L. braziliensis* isolate in the collection was genotyped at each *locus* according to their haplotype contents, considering diploid genomes. (A and C) Proportions of CTTCAG: CTTCAG (green), CTTCAG: TCATGA (red) and TCATGA: TCATGA (blue) in CHR24/3074 for the primary (A; i.e. 2008–2011) and validation (C; i.e. 1999–2001) parasite samples, respectively. (B and D) Proportions of CCT:CCT (green), CCT:TT- (red) and TT-:TT- (blue) in CHR28/425451 for the primary (B) and validation (D) samples, respectively. Overall fluctuations of genotype frequencies within each *locus* in each sample were not statistically significant (Chi-square p>0.05).

Furthermore, pairwise Cusick and Edward's comparisons between global distributions of heterozygous versus combined homozygous genotypes rendered non-significant (p>0.05) for both *loci* in the two study samples, indicating their ample overlap in that region and supporting HWE findings.

## Long-term stability of the *L. braziliensis* population in Corte de Pedra

The level of differentiation among subpopulations like those represented by the two samples in this study may be determined by the fixation index Fst, taking into account the heterozygosity indices $H_T$ and $H_S$. Fst explores the expected frequencies of heterozygotes (i.e. 2pq) in each sample being compared, and the overall frequencies observed for the involved alleles (i.e. p and q; CTTCAG and TCATGA for CHR24/3074; CCT and TT- for CHR28/425451) when the

**Table 2. Pairwise linkage disequilibrium between SNPs/indels in polymorphic positions of two *loci* located in chromosomes 24 (*locus* CHR24/3074) and 28 (*locus* CHR28/425451) of two temporally distinct *L. braziliensis* populations endemic in Corte de Pedra, Northeast Brazil: 1999–2001 (above gray diagonal), 2008–2011 (below gray diagonal).**

| *Locus* | | | *Locus* | | | | | | | | |
|---|---|---|---|---|---|---|---|---|---|---|---|
| | | | CHR 24 / 3074 | | | | | | CHR 28 / 425451 | | |
| *Locus* | CHR 24 3074 | **Positions** | **31** | **52** | **68** | **129** | **469** | **604** | **30** | **286** | **545** |
| | | **31** | | 0.0 | 0.0 | 0.0 | 0.0 | 0.0 | 0.4 | 0.4 | 0.4 |
| | | **52** | 0.0 | | 0.0 | 0.0 | 0.0 | 0.0 | 0.4 | 0.4 | 0.4 |
| | | **68** | 0.0 | 0.0 | | 0.0 | 0.0 | 0.0 | 0.4 | 0.4 | 0.4 |
| | | **129** | 0.0 | 0.0 | 0.0 | | 0.0 | 0.0 | 0.4 | 0.4 | 0.4 |
| | | **469** | 0.0 | 0.0 | 0.0 | 0.0 | | 0.0 | 0.4 | 0.4 | 0.4 |
| | | **604** | 0.0 | 0.0 | 0.0 | 0.0 | 0.0 | | 0.4 | 0.4 | 0.4 |
| | CHR 28 425451 | **30** | 0.1 | 0.1 | 0.1 | 0.1 | 0.1 | 0.1 | | 0.0 | 0.0 |
| | | **286** | 0.1 | 0.1 | 0.1 | 0.1 | 0.1 | 0.1 | 0.0 | | 0.0 |
| | | **545** | 0.1 | 0.1 | 0.1 | 0.1 | 0.1 | 0.1 | 0.0 | 0.0 | |

two samples are combined, according to the equation Fst = $(H_T - H_S)/H_T$. The expected heterozygote frequencies (2pq) at CHR24/3074 were 0.22 and 0.36, and the observed allele frequencies were 0.87 and 0.76 for CTTCAG (i.e. p) plus 0.13 and 0.24 for TCATGA (i.e. q) in the 1999–2001 and 2008–2011 samples, respectively. This resulted in a fixation index of Fst = 0.01, based on $H_T = 0.3345$ and $H_S = 0.3300$. For CHR28/425451, the expected heterozygote frequencies (2pq) were 0.49 and 0.49, and the observed allele frequencies were 0.44 and 0.58 for CCT (i.e. p) plus 0.55 and 0.41 for TT- (i.e. q) in the 1999–2001 and 2008–2011 samples, respectively. This resulted in a fixation index of Fst = 0.01, based on $H_T = 0.4951$ and $H_S = 0.4883$. Fixation indexes below 0.05 indicate very little to no genetic differentiation between two compared subpopulations, leading us to conclude that the *L. braziliensis* population in Corte de Pedra has been stable for at least one decade.

To further test for the existence of genetic differentiation among *L. braziliensis* in Corte de Pedra, we searched for indications of structuralization among subgroups of individuals in the two temporal samples using the population genetics tool Structure. We performed the analyses based on the two tested *loci* according to two parameters: presence of two putative subpopulations in the dataset (K = 2), and presence of twenty subpopulations in the dataset (K = 20). Independent of the condition established each individual of the dataset formed by the two temporal samples (i.e. 1999–2001 and 2008–2011) presented the same contribution to theoretical subpopulations. When K = 2, each individual presented approximately 50% contribution to each subpopulation (Fig 5A). When K = 20, each *L. braziliensis* isolate contributed 5% to each theoretical subpopulation (Fig 5B). Thus, no structuralization could be found, further reinforcing the findings for Fixation indices described above.

## Discussion

*Leishmania* spp. involved in natural infections of human beings have been amply described in the literature according to various parameters of their clinical isolates like reactivity to antibody, isoenzyme content and genetic profile. In general, these parasites have shown to be remarkably polymorphic with some species standing out, like *L. braziliensis* that has been shown to present several serologic [23], phenotypic [24–26] and genetic [27,28] variant types. Despite of all the knowledge accumulated, it is still not completely understood how *L. braziliensis* and other species of the parasite multiply and segregate their genetic material to the offspring generations within foci endemic for leishmaniasis.

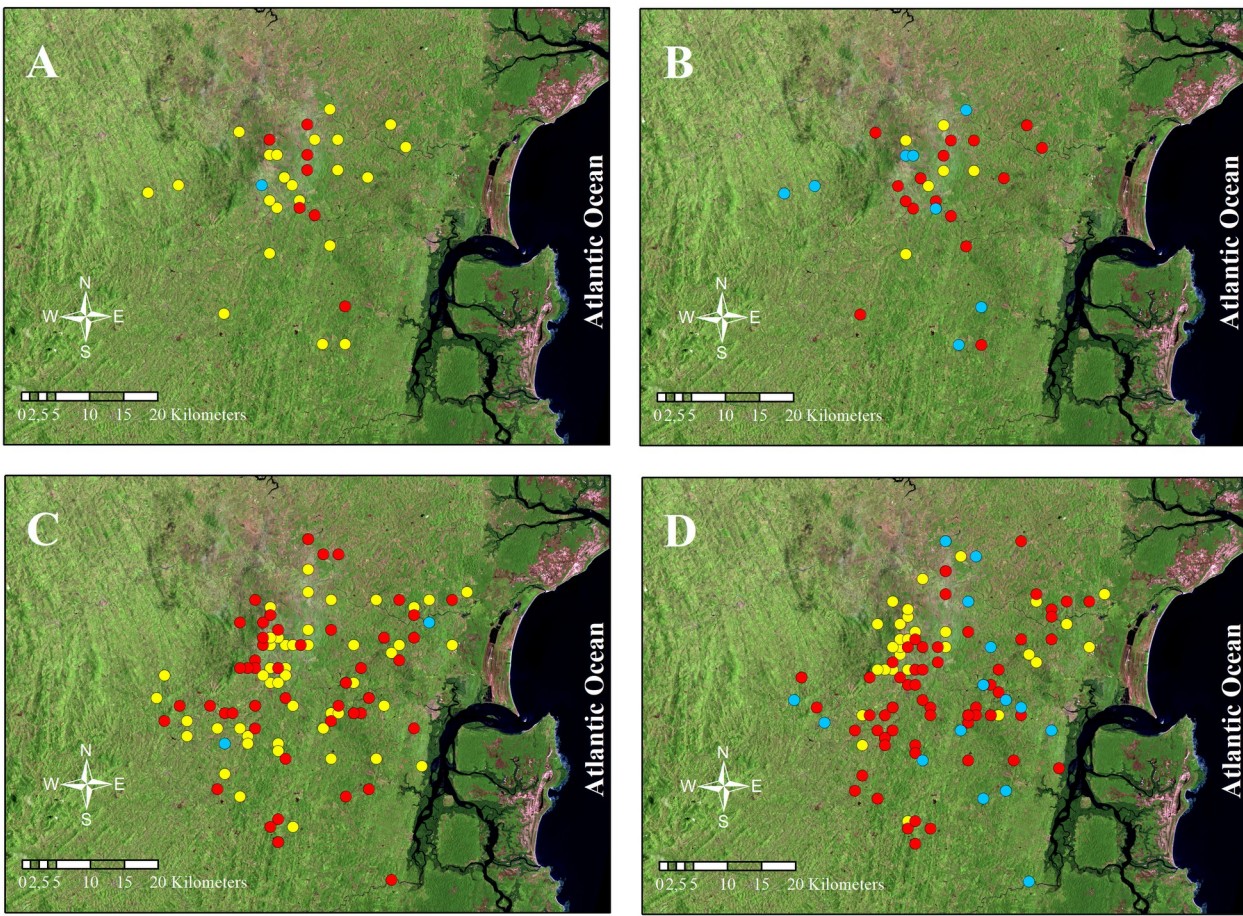

**Fig 4. Homozygous and heterozygous genotypes in *L. braziliensis* loci greatly overlap in Corte de Pedra.** Homozygous and heterozygous genotypes observed in *L. braziliensis* loci CHR24/3074 and CHR28/425451 of parasites obtained during 1999–2001 (A and B) and 2008–2011 (C and D) sampling periods were mapped and the resulting sets of geographic events were statistically compared. CHR24/3074 (A and C): CTTCAG: CTTCAG (yellow), CTTCAG: TCATGA (red) and TCATGA: TCATGA (blue). CHR28/425451 (B and D): CCT:CCT (yellow), CCT:TT- (red) and TT-:TT- (blue). Cuzick and Edward´s comparisons of combined homozygous versus heterozygous spatial distributions rendered non-significant (p>0.05) for data depicted in all four maps of the panel. Total number of dots plotted in each map may be smaller than the number of corresponding parasite isolates described in text due to overlap of some ATL patients' geographic coordinates. Link to the United States Geological Survey (USGS) Landsat satellite photograph used in the figure [39]: https://earthexplorer.usgs.gov/scene/metadata/full/5e83d1193824e4fc/LT52160691994219CUB00/.

The longstanding observations that both promastigotes and amastigotes divide by binary fission in vitro has strongly supported the vision that *Leishmania* spp. could probably maintain their natural populations clonally multiplying. Sound early molecular epidemiological studies, often based on the combination of electrophoretic profiles or limited DNA sequence data with phenetic or cladistic classification of generous samples of strains, helped further sediment this conclusion [10,29,30]. However, many of those studies highlighted that exceptions might occur, often detectable as hybrid individuals presenting typing patterns consistent with those of two distinct species within a single strain of the parasite [10,31–35]. The general conclusion was that *Leishmania* spp. populations were maintained mostly asexually, while bouts of sexual reproduction leading to the hybrids might eventually take place.

Recent experimental research has shaken the dogma showing that *L. major* and possibly other species of *Leishmania* are indeed capable of mating and leading to hybrid offspring

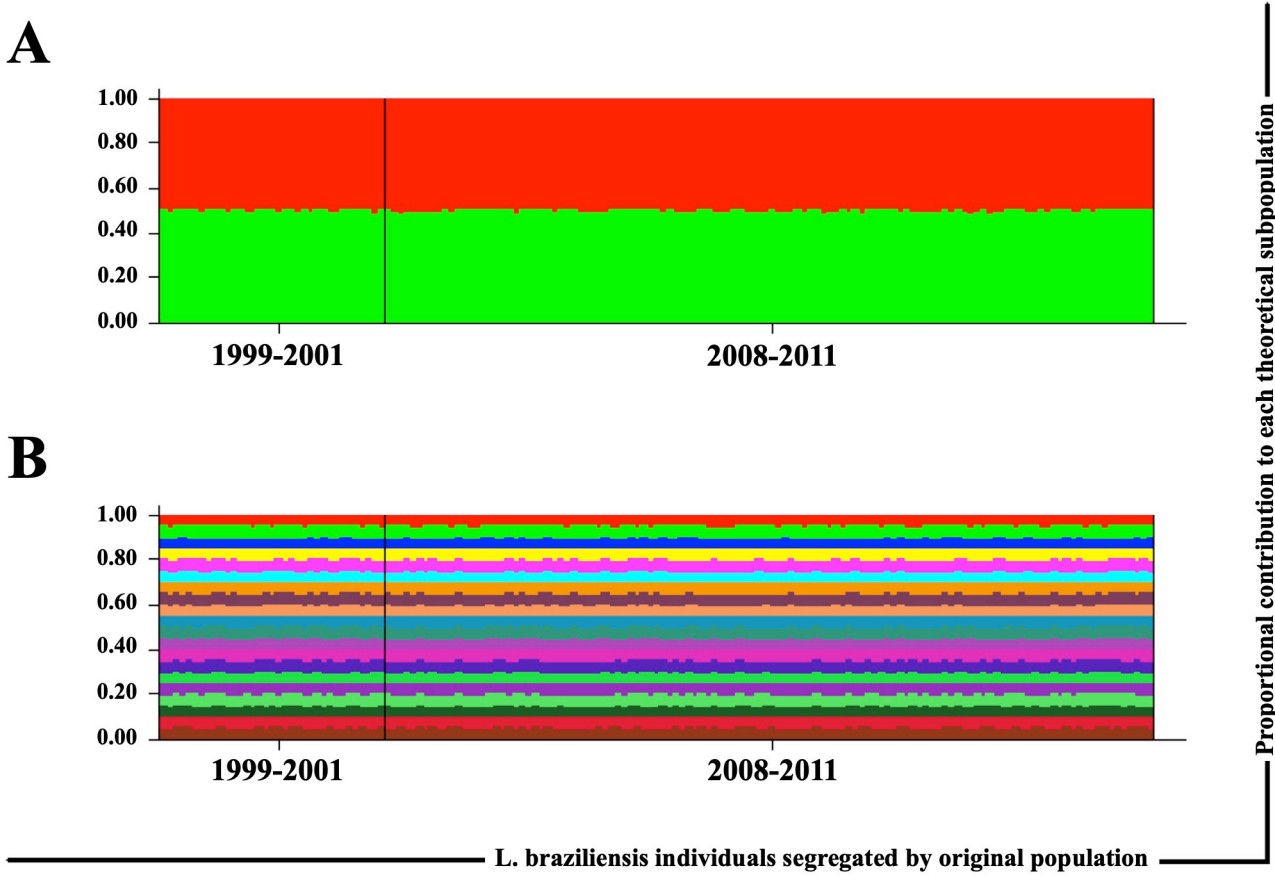

**Fig 5. *L. braziliensis* individuals contribute similarly to putative subpopulations predicted by Structure, based on their genotypes in CHR24/3074 and CHR28/425451 *loci*, in Corte de Pedra.** Structuralization within Corte de Pedra's *L. braziliensis* population into discrete subpopulations was assessed using the software STRUCTURE [21]. The analyses were performed using combined data of CHR24/3074 and CHR28/425451 *loci*. Two conditions were tested: (A) structuralization into two subpopulations, i.e. K = 2; and (B) structuralization into 20 subpopulations, i.e. K = 20. Histograms show the proportional contributions of each individual in the overall sample to each theoretical subpopulation ordered by sampling period (i.e. 1999–2001 and 2008–2011) after a burn-in step of 10,000 runs followed by 100,000 MCMC replications of the data.

within sand flies [11,12]. These reports suggest that both sexual and asexual reproduction may be relevant to the life cycle of the parasites albeit occurring within distinct hosts in natural populations of the protozoa. Good examples of human disease-causing protozoa that share this reproduction pattern are the *Plasmodium* spp. They multiply asexually by schizogony within human hosts and sexually during sporogony in the invertebrate hosts.

Our findings for parasites isolated from patients in one of the regions with highest endemicity for ATL in Brazil support this hypothesis. In particular, the observations that homozygous and heterozygous genotypes in two different *L. braziliensis* chromosomes are in Hardy-Weinberg equilibrium and are not in linkage disequilibrium. They indicate that genetic recombination within that natural population is robust, possibly due to frequent sexual reproduction events among *L. braziliensis* individuals within invertebrate and / or vertebrate hosts during their life cycles. We acknowledge, though, that the evaluation of just two small *loci* may not entirely represent a genome of greater than 30 M bases distributed in more than 30 chromosomes. Future work exploring whole genome sequences is in order to further dissect whether frequent genetic recombination homogeneously affects the entire genome of the parasite.

A pioneering study had previously assessed the population genetics of *L. braziliensis*, exploring a large panel of micro-satellite markers and F statistics [36]. The research included individuals from four populations located in the Amazon, sampled from one focus of ATL in Peru and one in Bolivia, spanning about 100 to 200 Km$^2$ each and yielding 124 parasite isolates. Its major conclusions were that sex-like reproduction with intense inbreeding among strains of the parasite occurred in those populations. The authors also found a suppression in heterozygous genotypes in comparison with what would be expected by Hardy-Weinberg prediction, and evidence of strong structuralization reflected in the observation of multiple clusters of individuals in the samples analyzed. They inferred that these clusters might be consequences of the existence of several different ecotypes in the area of sampling within the highly diversified flora and fauna of the rain forest.

In the current study, we revisited the population genetics of *L. braziliensis* in a rural setting hyperendemic for ATL that spans for 10,000 Km$^2$, close to the Atlantic Ocean in Northeast Brazil. We employed two sets of haplotypes of six and three SNPs or indels that marked stretches of precisely 573 and 515 nucleotides (i.e., measured from first to last SNP / indel in the locus) in chromosomes 24 and 28 of the parasites, respectively. Overall, we detected that these two loci were in HWE, reinforcing the conclusion on the existence of sex-like reproduction in part of the *L. braziliensis* life-cycle made by Rougeron et al. [36]. However, we did not find evidence of heterozygous suppression or structuralization in that population.

We believe that the partial discrepancy in these observations is not related to the different molecular and statistical approaches of the two studies. We think that the major driver for the partially contrasting observions is the difference between the highly diversified ecosystem of the Amazon rain forest represented in Rougeron et al. [36] and the much less heterogeneous fauna and flora of a rural area dominated by farms with residual forest coverage and small to mid-sized villages in the current study. It is also likely that human beings dwelling in this ATL hyperendemic region acquire the parasites both while working within the residual forests, used to grow cocoa crops, as well as in the vicinities of their homes, what coupled to the intense day-to-day movement of these individuals within and between villages of the region may cause the more even distributions and prevent the clustering of genotypes, as we detected in the current study. All in all, we believe that the two studies offer insights of complementary realms that coexist in the transmission cycles of *Leishmania* spp.

The fixation indices (Fst) and structuralization data obtained for parasite samples collected 10 years apart indicate that the study *L. braziliensis* population is capable of maintaining its stability for long periods. Nevertheless, time distribution of genotypes also showed that some variation in their frequencies occur between transmission seasons. This is consistent with our previous description of the dynamics of the ATL in Corte de Pedra. In that region, the human endemics is maintained by multiple outbreaks of the disease at each transmission season [16,17]. Given the heterogeneity detected in that population during current and past studies [5,6,15], these concurrent and successive outbreaks of ATL may be accompanied by a random success of different strains of the parasite to reach the human population in the affected focus at each transmission season. This would translate into the time fluctuation in the genotype frequencies observed for CHR24/3074 and CHR28/425451.

It is important to point out that the *L. braziliensis* samples in the study represent the fraction of the overall parasite population that effectively reaches human beings and cause clinically evident infections in Corte de Pedra. Certainly, a less chance sensitive approach would be to actively surveil for and isolate parasites from symptomatic and asymptomatic humans as well as non-human hosts and reservoirs of *L. braziliensis* in the focus of endemicity. We speculate, though, that such approach would ultimately help reinforce our findings on the HWE, LD

and genetic differentiation in this population by increasing the opportunity for less frequent genotypes like CHR24/3074 TCATGA homozygotes to be better represented in the samples.

Another source of limitation in our approach stemmed from using DNA from freshly isolated *L. braziliensis* in the primary study sample (i.e. enrolled in 2008–2011) and first passage cultured parasites in the validation sample (i.e. obtained in 1999–2001). Our option aimed at employing in our analyses groups of individuals that could best represent the natural population circulating in Corte de Pedra, avoiding as much as possible any bias that multiple passages into cultures may cause to isolates. The alternative would be to use cloned parasites from each isolate. Such approach would allow for unambiguous genotyping of heterozygotes, discriminating true heterozygotes from mixed infection isolates consisting of two different homozygotes. However, this strategy would be too laborious to be unmistakably carried out with the fairly numerous isolates used in this study. Besides, we think that the redundancy in the findings obtained for different *loci* and samples, using complementary analytical approaches helped alleviate the consequences of chance in our conclusions.

We opted to use haplotypes of polymorphic nucleotides in biallelic positions at CHR24/3074 and CHR28/425451 to increase the precision of our HWE and LD analyses. This approach helped the consistent genotyping of isolates, revealing the existence of predominant and rare haplotypes at each tested *locus* in the *L. braziliensis* of Corte de Pedra. Interestingly, a few isolates concomitantly displayed a heterozygous CTTCAG:TCATGA genotype in the presence of one of the rare haplotypes in CHR24/3074. This suggests that some level of mixed *L. braziliensis* strains infections may indeed occur in inhabitants of that region.

We cannot rule out that the use of other markers we had previously detected in other chromosomes of the parasite [5] might have led to different outcomes, namely that part or all of them would not present Hardy Weinberg Equilibrium and present linkage disequilibrium in the study *L. braziliensis* population. It is important to point out that the approach we opted for was very conservative, consisting in culturing over 150 isolates of *L. braziliensis*, then cloning over 300 inserts into plasmid vectors and finally sequencing over 1,800 clones. So, it ended up neither practical nor cost effective for us to expand the analyses to the other four loci we had previously identified [5]. We compromised by opting for two loci that located in two different chromosomes and that were both marking putative coding regions previously annotated in the *L. braziliensis* genome.

The current study did not attempt to give a definitive answer as to the modes of reproduction of *L. braziliensis*. We believe that an extensive genome-wide approach is necessary to appropriately address this issue, given particularities of Trypanosomatids like the organization and expression of their genomes. In the current status of such knowledge, though, we think this research helps reinforce the amounting evidence that sex-like reproduction seems to occur at some point of the parasite's life cycle in nature [11,12,36].

Our ultimate focus is on the practical consequences that frequent genetic exchange may present for the management of the endemics. Particularly regarding predictive diagnosis. Patients of ATL may evolve to severe forms of disease and present refractoriness to some of the drugs used to treat them. So, the development of diagnostic protocols capable of predicting these outcomes is most welcome in the field. Although we do not envision CHR24/3074 and CHR28/425451 use for diagnosis, our findings that some chromosomal markers may be in Hardy Weinberg Equilibrium in a population of *L. braziliensis* highlight the utmost importance of thoroughly testing the short- and long-term accuracy of predictive diagnostic tools based on nucleic acid detection before their roll out to the ATL affected foci.

Chromosomal markers are inherited vertically by virtually all individuals of a given strain in microbial species that multiply clonally, enabling their precise identification within the overall population. This is opposed to the realm of sexually multiplying organisms, for which a

region of the genome can only consistently mark a trait if it is physically encoded in disequilibrium with the gene or genes responsible for the targeted phenotype. Leishmaniases are neglected tropical diseases that cause a large burden [2] to ten percent of the world's population, who live in regions at risk in some of the poorest areas of the globe. These disorders may cause an array of outcomes in patients, substantially varying in their severity [37,38]. For these reasons, the search of biomarkers that could prognosticate clinical progression and therapeutic success consists in a field of intensive research in leishmaniasis. The dwellers of affected foci often have limited access to health care and would greatly benefit from predictive diagnostic tools that might help effectively stratify them according to parameters like probable treatment response, odds of aggressive disease and chance of long-term recurrence that would require periodic follow-up. Such mostly needed translational research would profoundly benefit of and further flourish with the accurate understanding of the population genetics of *Leishmania* spp.

## Acknowledgments

We deeply thank all personnel of the Jackson Costa Health Post in Corte de Pedra, Bahia, Brazil, for their careful help with patient management.

## Author Contributions

**Conceptualization:** Mary E. Wilson, Albert Schriefer.

**Data curation:** Albert Schriefer.

**Formal analysis:** Juliana A. Silva, Ana Isabelle Pinheiro, Maria Luiza Dourado, Albert Schriefer.

**Funding acquisition:** Mary E. Wilson, Edgar M. Carvalho, Albert Schriefer.

**Investigation:** Juliana A. Silva, Ana Isabelle Pinheiro, Maria Luiza Dourado, Lilian Medina, Adriano Queiroz, Luiz Henrique Guimarães, Marcus Miranda Lessa, Ednaldo L. Lago, Paulo Roberto L. Machado, Edgar M. Carvalho, Albert Schriefer.

**Methodology:** Albert Schriefer.

**Writing – original draft:** Albert Schriefer.

**Writing – review & editing:** Albert Schriefer.

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
