## [Decision Letter · Decision Letter 0]

29 Jun 2021

Dear Dr. Schriefer,

Thank you very much for submitting your manuscript "Leishmania braziliensis causing human disease in Northeast Brazil presents loci with genotypes in long-term equilibrium." for consideration at PLOS Neglected Tropical Diseases. As with all papers reviewed by the journal, your manuscript was reviewed by members of the editorial board and by several independent reviewers. In light of the reviews (below this email), we would like to invite the resubmission of a significantly-revised version that takes into account the reviewers' comments. 

We cannot make any decision about publication until we have seen the revised manuscript and your response to the reviewers' comments. Your revised manuscript is also likely to be sent to reviewers for further evaluation.

Sincerely,

Abhay R Satoskar

Deputy Editor

Abhay Satoskar

Deputy Editor

Reviewer's Responses to Questions

**Key Review Criteria Required for Acceptance?**

**Methods**

-Are the objectives of the study clearly articulated with a clear testable hypothesis stated?

-Is the study design appropriate to address the stated objectives?

-Is the population clearly described and appropriate for the hypothesis being tested?

-Is the sample size sufficient to ensure adequate power to address the hypothesis being tested?

-Were correct statistical analysis used to support conclusions?

-Are there concerns about ethical or regulatory requirements being met?

Reviewer #1: The paper was well written and the text is very clear, allowing publication without revision of the english language. The objectives of the study is clearly articulated with a clear testable answers and the study design is appropriate to address the stated objectives.

The study population is especially important because it is a community village in the northeast region of Brazil, an extremely vulnerable population and an important endemic area of ATL. The study population is clearly described and appropriate for the hypothesis being tested.

I emphasize that the sample size used in this work is very substantial and sufficient to ensure adequate power to address the hypothesis being tested, which further reinforces the quality of the work.

However I think that the loci selected for the study might be too small region/sequence to investigate if there is sex-like recombination in a complex population of Leishmania. This would be an important bias of the work.

The authors need to comment further on how the molecular targets were selected for this work. And also discuss this possible bias.

The statistical analysis were correctly used to support conclusions and all ethical requirements have been met.

Moreover, presented a methodology suitable for evaluating these purposed aims; the results allowed a very detailed argumentation of the data in a holistic way.

Reviewer #2: How do the results compare with previously published studies on L braziliensis that examined microsatellite loci to investigate linkage disequilibrium in Peru and Bolivia [Rougeron et al, PNAS 2009]? How are the ~600nt genetic loci on chromosomes 24 and 28 likely to reveal new insights? 

Is the lack of linkage disequilibrium between the parasite samples due to the targets selected for investigation? Could inclusion of other potential targets as described in the previous studies bu the authors changed the outcome?

Are the temporally distinct sample sets collected almost a decade apart have gone through a number of in vitro replication cycles. How did the authors ensure the validity of the results? Was it done by random assignment of the testing set and validation sets of samples?

Are there any current diagnostics assays based on the genetic loci from chromosomes 24 and 28 or the authors are making a more generalized statement with respect to the diversity of the loci studied. The authors need to make a clear case for the utility of the findings of the study.

**Results**

-Does the analysis presented match the analysis plan?

-Are the results clearly and completely presented?

-Are the figures (Tables, Images) of sufficient quality for clarity?

Reviewer #1: Yes, the analyzes corresponded to the experimental design proposed.

The results topic is well organized into subtopics. It only needs clarification on some points that will be pointed out in the document requesting a minor revision.

In relation to the figures I pointed out a modification suggestion in Figure 2 to bring a clearer understanding.

Reviewer #2: The results are clearly presented.

**Conclusions**

-Are the conclusions supported by the data presented?

-Are the limitations of analysis clearly described?

-Do the authors discuss how these data can be helpful to advance our understanding of the topic under study?

-Is public health relevance addressed?

Reviewer #1: The conclusions are supported by the data presented except for the statement that the population is in HWE in contradiction to saying that there is gene flow and reproduction as sexual. I comment on this in article review requests. The limitations of the analyzes and the interpretation bias of the results are clearly discussed by the authors. Only a few minor adjustments are needed. Surely, the paper have a public relevance addressed.

Reviewer #2: The conclusions are supported by the data presented. See my comments above

**Editorial and Data Presentation Modifications?**

Reviewer #1: The following are the reviews and questions that must be presented by the authors before publication:

1. Lines 114-117: I think this might be too small a region/sequence to investigate if there is sex-like recombination in a complex population of Leishmania. This would be an important bias of the work. In the methodology, the authors need to comment further on how the molecular targets were selected for this work. And also discuss this possible bias.

2. Lines 123-130: Please add the related reference to the information presented about the study area.

3. Lines 326-328: I suggest that the authors explain here which values would indicate the existence of LD and which values are assumed to be absence of LD.

4. Lines 397-399: I suggest adding more recent articles on this topic: Molecular tools confirm natural Leishmania (Viannia) guyanensis/L. (V.) shawi hybrids causing cutaneous leishmaniasis in the Amazon region of Brazil (Lima et. al., 2021); Phenotypic characterization of Leishmania spp. causing cutaneous leishmaniasis in the lower Amazon region, western Pará state, Brazil, reveals a putative hybrid parasite, Leishmania (Viannia) guyanensis × Leishmania (Viannia) shawi shawi (Jennings et. al., 2014).

5. Lines 404-410: One of the most important assumptions of the Hardy-Weinberg equilibrium is the absence of gene flow. This disagrees with his statement that sexual and asexual reproduction may be equally relevant to the life cycle of the parasites and that the findings for parasites isolated from patients in this study support this hypothesis. Since it is not possible for a population to be in HWE and at the same time have sexual reproduction, I ask the authors to clarify better this statement further.

6. Lines 654-664 / Figure 2: In my opinion this figure is not the best way to demonstrate the nucleotide haplotype data found in the two chromosome loci studied. I suggest using the cluster dendrogram presentation, or that the authors propose another possibility to show these results.

Reviewer #2: (No Response)

**Summary and General Comments**

Reviewer #1: (No Response)

Reviewer #2: The authors examined the genetic characteristics of L braziliensis parasite isolates collected over a 10 year period from a region hyper-endemic for ATL. Based on their previous studies, they selected two chromosomal loci for further examination. Sequencing and analysis of the 600nt segments from chromosomes 24 and 28 revealed no linkage disequilibrium thus suggesting that L braziliensis can maintain stable populations over time.

PLOS authors have the option to publish the peer review history of their article (what does this mean?). If published, this will include your full peer review and any attached files.

Reviewer #1: Yes: Patricia Flávia Quaresma

Reviewer #2: No
---

## [Decision Letter · Decision Letter 1]

3 Apr 2022

Dear Dr. Schriefer,

We are pleased to inform you that your manuscript 'Leishmania braziliensis causing human disease in Northeast Brazil presents loci with genotypes in long-term equilibrium.' has been provisionally accepted for publication in PLOS Neglected Tropical Diseases.

Best regards,

Abhay R Satoskar

Deputy Editor

Abhay Satoskar

Deputy Editor

Reviewer's Responses to Questions

**Key Review Criteria Required for Acceptance?**

**Methods**

-Are the objectives of the study clearly articulated with a clear testable hypothesis stated?

-Is the study design appropriate to address the stated objectives?

-Is the population clearly described and appropriate for the hypothesis being tested?

-Is the sample size sufficient to ensure adequate power to address the hypothesis being tested?

-Were correct statistical analysis used to support conclusions?

-Are there concerns about ethical or regulatory requirements being met?

Reviewer #1: Considering that the manuscript contain a significant information to justify publication, that the problem was significant and concisely stated, the methods are described comprehensively and the interpretations and conclusions were justified by the results, I recommend the article for publication.

**Results**

-Does the analysis presented match the analysis plan?

-Are the results clearly and completely presented?

-Are the figures (Tables, Images) of sufficient quality for clarity?

Reviewer #1: All adjustments requested to improve the presentation of results were duly carried out by the authors. The results are clearly and completely presented.

**Conclusions**

-Are the conclusions supported by the data presented?

-Are the limitations of analysis clearly described?

-Do the authors discuss how these data can be helpful to advance our understanding of the topic under study?

-Is public health relevance addressed?

Reviewer #1: I thank the authors for made all the clarifications and improvements I requested in the Discussion and Conclusions items. Now the text is much better for publication.

**Editorial and Data Presentation Modifications?**

Reviewer #1: Dear Editor,

Considering that all the questions raised by me were duly clarified and that the suggestions and corrections pointed out to improve the text were also met, I affirm that the revised version of the article is approved and may be published.

**Summary and General Comments**

Reviewer #1: Dear authors,

Considering that all the questions raised by me were duly clarified and that the suggestions and corrections pointed out to improve the text were also met, I affirm that the revised version of the article is approved and may be published. Congratulations for the hard and important work that brings an enormous contribution to the scientific area of studies on leishmaniasis.

PLOS authors have the option to publish the peer review history of their article (what does this mean?). If published, this will include your full peer review and any attached files.

Reviewer #1: **Yes: **Patricia F Quaresma

---

## [Editor Report · Acceptance letter]

10 Jun 2022

Dear Dr. Schriefer,

We are delighted to inform you that your manuscript, "Leishmania braziliensis causing human disease in Northeast Brazil presents loci with genotypes in long-term equilibrium.," has been formally accepted for publication in PLOS Neglected Tropical Diseases.

Best regards,

Shaden Kamhawi

co-Editor-in-Chief

Paul Brindley

co-Editor-in-Chief
